DATA RELEASE

# The major and minor *Aedes* mosquitoes from southern Nigeria exhibit low resistance towards public health insecticides

Udoka C. Nwangwu[1,*], Patience O. Ubachukwu[2], Peter C. Okeke[2,3], Muhammad M. Mukhtar[4], Chukwuebuka M. Nwosu[1], Ifeoma M. Ngwu[1], Oscar N. Nwaogo[1], Stephen O. Anokwu[1], Linda C. Ikechukwu[1], John E. Ogbu[1], Nneka O. Agashi[1], Chukwuebuka K. Ezihe[5], Festus A. Dogunro[1], Cosmas O. Onwude[1], Emelda I. Eloy[1], Ijeoma U. Ikeakor[6], Emmanuel O. Nwosu[1], Spencer C. Nwangwu[7], Chiamaka U. Nwangwu[8], Joseph U. Anumba[9], Razaki A. Osse[10], Arthur Sovi[10,11,12], Fiacre R. Agossa[10], Chukwuemeka C. Asadu[1], Okechukwu C. Chukwuekezie[1] and Sulaiman S. Ibrahim[4,13,*]

1 National Arbovirus and Vectors Research Centre (NAVRC), #33 Park Avenue, GRA, Enugu State, Nigeria
2 Department of Zoology and Environmental Biology, P.O. Box 3, University of Nigeria, Nsukka, Enugu State, Nigeria
3 Opec Research Consult, #1 Uche Ekwunife Road, Awka, Anambra State, Nigeria
4 Department of Biochemistry, Bayero University, PMB 3011, Kano State, Nigeria
5 Malaria Consortium, #33 Pope John Paul Street, Off Gana Street, Abuja, Nigeria
6 University of the Western Cape, Robert Sobukwe Road Bellville 7535, Cape Town, South Africa
7 Department of Biochemistry, Igbinedion University, P.M.B 0006, Okada, Edo State, Nigeria
8 Nigeria Centre for Disease Control (NCDC), Plot 801, Ebitu Ukiwe Street, Jabi, Abuja, Nigeria
9 Health Research Institute, University of Canberra, Bruce, Australian Capital Territory, Australia
10 Centre de Recherche Entomologique de Cotonou (CREC), 06BP2604 Cotonou, Bénin
11 London School of Hygiene and Tropical Medicine, London, UK
12 Faculty of Agronomy, University of Parakou, Parakou, Benin
13 Centre for Research in Infectious Diseases (CRID), P.O. Box 13591, Yaoundé, Cameroon

**Submitted:** 24 August 2025

* Corresponding authors. E-mail:
nwangwu_udoka@navrc.org.ng;
ssibrahim.bch@buk.edu.ng

Preprint submitted at
https://africarxiv.ubuntunet.net/handle/1/10395

Included in the series: *Vectors of human disease* (https://doi.org/10.46471/GIGABYTE_SERIES_0002)

## ABSTRACT

Insecticide-based interventions continue to serve as the cornerstone of *Aedes* mosquito control, the primary vectors of arboviruses. This study assessed the insecticide resistance profiles of four *Aedes* mosquitoes in three rural areas in southern Nigeria, where arbovirus outbreaks recently occurred. Using WHO tube tests and CDC bottle bioassays, four *Aedes* species (*Aedes aegypti*, *Ae. albopictus*, *Ae. simpsoni* complex and *Ae. luteocephalus*) were evaluated for susceptibility to commonly used public health insecticides, including deltamethrin, alphacypermethrin, permethrin, pirimiphos-methyl, chlorfenapyr and clothianidin. Biochemical assays were conducted using *Ae. albopictus* to establish the role of metabolic resistance mechanism. Amplification and sequencing of fragment of *Ae. luteocephalus* ITS1 gene molecularly confirmed its species identity. *Aedes aegypti* exhibited possible resistance to pirimiphos-methyl but remained susceptible to all other insecticides across study sites. *Aedes albopictus* showed resistance to DDT and possible resistance to pirimiphos-methyl, while remaining susceptible to pyrethroids. *Aedes luteocephalus* was resistant to pirimiphos-methyl but susceptible to all other insecticides. *Aedes simpsoni* complex was fully susceptible to all insecticides. Biochemical assays revealed elevated $\alpha$-esterase and monooxygenase activities (3.4-fold and 2.54-fold, respectively) in exposed females of *Ae. albopictus* compared to the unexposed cohort. Overall, the low

resistance levels observed underscore the need for sustained insecticide resistance monitoring and management to maintain the effectiveness of insecticide-based vector control strategies in Nigeria.

**Subjects**  Ecology, Biodiversity, Taxonomy

## DATA DESCRIPTION
## Background and context

The global threat posed by arboviruses transmitted by *Aedes* mosquitoes has intensified in recent years, prompting the World Health Organization (WHO) to launch the Global Arbovirus Initiative, cautioning that the next pandemic could be an arbovirus [1]. Increasing reports of arboviral outbreaks have emerged across the world, including in Europe, where autochthonous transmission of dengue, chikungunya, and Zika has been documented [2, 3]. In Brazil, dengue infections surged by 108% between 2023 and 2024 [4], while Bangladesh recorded its deadliest dengue outbreak in 2023, following the resurgence of Dengue virus serotype 2 [5]. Similarly, Sri Lanka reported major outbreaks caused by co-circulating genotypes I and III of dengue virus serotype 3 [6].

In Africa, arbovirus outbreaks have also increased in frequency and intensity. In 2023 alone, seven arboviruses were implicated in 29 outbreaks across 25 countries, resulting in 19,569 confirmed cases and 820 deaths. Notably, West Africa accounted for 22 of these outbreaks, including those in Senegal, Burkina Faso, Côte d'Ivoire, Ghana, Togo, and Nigeria [7].

*Aedes aegypti* and *Ae. albopictus*, the primary vectors of yellow fever, dengue, chikungunya, and Zika viruses, are expanding their geographical range and have adapted to a wide array of habitats [8–12]. These mosquitoes thrive in artificial containers in urban settings and increasingly use natural containers such as plant axils and tree holes in rural areas, thereby enhancing their transmission potential across ecotypes. Furthermore, insecticide resistance in *Aedes* mosquitoes is emerging as a major public health concern, threatening the efficacy of core vector control strategies globally [13, 14].

Resistance to the four major insecticide classes (pyrethroids, organophosphates, carbamates, and organochlorines) has been reported in *Ae. aegypti* and *Ae. albopictus* across the Americas, Asia, and Africa [15]. This resistance is largely attributed to target-site mutations (such as knockdown resistance or *kdr* mutations) and enhanced metabolic detoxification, including elevated activity of glutathione S-transferases (GSTs) and cytochrome P450 monooxygenases [16–19]. The reduced efficacy of space spraying and insecticide-treated materials leads to increased vector survival, limiting outbreak response effectiveness and posing a serious challenge to arbovirus prevention [20, 21]. Across Africa, several studies have documented varying resistance profiles in *Aedes* mosquitoes, often linked to urbanization and insecticide pressure, underscoring the need for context-specific resistance management strategies [11, 14, 22].

In Nigeria, *Aedes*-borne arboviruses, such as yellow fever [23], dengue [24] and chikungunya [25], continue to cause recurrent outbreaks. As of July 2025, Nigeria had reported suspected yellow fever cases across the country, resulting in seven confirmed cases in six states (Abia, Anambra, Edo, Ekiti, Lagos, and Rivers), while a large outbreak of

dengue was ongoing in Edo State [26]. The distribution of *Aedes* species varies by ecological zone, with *Ae. aegypti* more prevalent in urban environments, while *Ae. albopictus* is consolidating its presence in cities and expanding into peri-urban and rural areas [27, 28]. Southern Nigeria harbours diverse *Aedes* species, including *Ae. luteocephalus*, *Ae. africanus*, *Ae. vittatus*, and *Ae. simpsoni* complex, which may play roles in localized arbovirus transmission [28, 29]. In contrast, only *Ae aegypti* has been reported in the Sahel and Sudan savannahs of the far north [12, 28].

Several studies have reported contrasting patterns in the distribution of major and minor *Aedes* species [28, 30] and resistance profiles across the 5 contrasting ecological zones of Nigeria. Understanding the geographic distribution, species composition, and resistance status of *Aedes* species, especially in understudied rural areas, is essential for designing and implementing evidence-based vector control strategies.

Several studies have revealed regional differences in insecticide resistance in *Aedes* species across Nigeria. In the humid southern regions, higher levels of multiple insecticide resistance have been reported in *Ae. aegypti* from the Southwest towards DDT, pyrethroids and carbamates [31, 32]. The same vector has recorded limited resistance to pyrethroid in the South-south [33], while DDT resistance in both *Ae. aegypti* and *Ae. albopictus* was recorded in the Southeast [34]. The resistance has been associated with the voltage-gated sodium channel (*vgsc*) knockdown resistance (*kdr*) mutations (F1534C, S989P, V1016G) and elevated detoxification enzymes [35]. Also, carbamate resistance has been recorded against *Ae. aegypti* in the North Central region [36], while in Northwestern Nigeria, its populations exhibited resistance towards organochlorine, organophosphate, carbamate and pyrethroid insecticides. The pyrethroid resistance was shown to be associated with the F1534C mutation and high levels of GSTs and cytochrome P450s, while V1016G mutation was absent [12].

Despite the studies mentioned above, data on insecticide resistance in rural communities remains largely limited. This is regardless of the increasing adaptation of *Aedes* mosquitoes to these areas. This gap hampers the design of targeted and effective control interventions. Moreover, while metabolic and target-site resistance mechanisms have been described in urban populations, their prevalence and impact in rural *Aedes* populations are poorly understood [11, 22, 31].

To address these gaps, this study investigated the composition and resistance profiles of *Aedes* species in rural areas of South-Eastern Nigeria. Four *Aedes* species were identified to species level, and their resistance towards public health insecticides was established to be very low. These findings not only fill a critical knowledge gap on rural *Aedes* populations in Nigeria but also highlight an important window of opportunity to implement proactive, evidence-based vector control strategies before resistance emerges. More of this resistance data needs to be made publicly available and linked to species and occurrence observations in the Global Biodiversity Information Facility (GBIF) database to enable better modelling and global control efforts.

## METHODS

### Mosquito sampling, rearing and morphological identification

Immature stages of *Aedes* mosquitoes were collected in July 2021 from three communities: Oyege (06.56328°N, 008.24153°E); Ndiezoke (06.55399°N, 008.23435°E) and Offerekpe



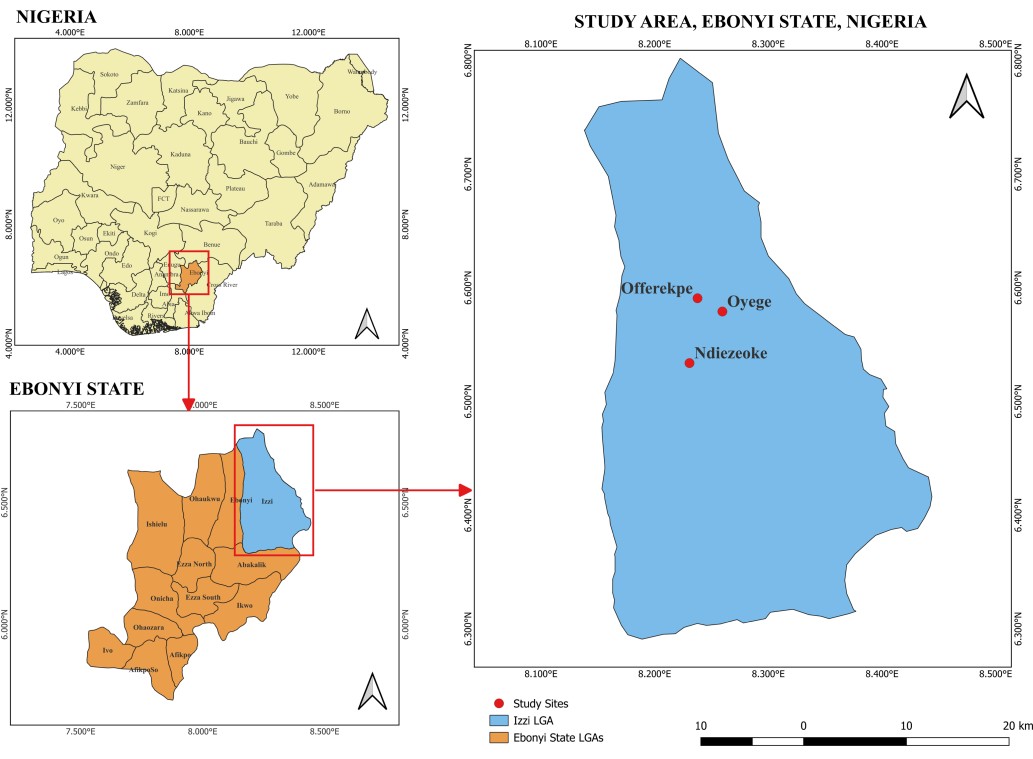

**Figure 1.** Map of the study area, showing sampling locations.

Inyimagu (06.57017°N, 008.23528°E), all within Izzi Local Government Area of Ebonyi State, Nigeria (Figure 1).

Larvae and pupae were collected from natural and artificial containers using ladles and pipettes, as described previously [28, 37]. The natural containers were mostly plant axils, while the artificial containers were mainly household water storage and discarded containers around human dwellings. Ovitraps, which were set and retrieved within 72 h, were used to augment collections. The immature stages collected from the field were raised to adulthood, anaesthetized and identified morphologically using the keys of [38] and [39]. All the field-collected mosquitoes were separated into distinct populations at the insectary of the National Arbovirus and Vectors Research Centre (NAVRC), Nigeria. Where the population was insufficient for bioassay, they were blood-fed to get F1 progenies, which were fed 10% sucrose, maintained at standard insectary conditions of temperature 27 °C ± 2, relative humidity of 80% ± 10 and a 12:12 hours light-dark cycle. The mosquitoes were used for bioassays. They were eventually preserved in Eppendorf tubes and stored at −20 °C in an ultra-low freezer for further analyses.

## Molecular identification of *Ae. luteocephalus* to species level

As *Ae. luteocephalus* showed interesting resistance toward pirimiphos-methyl in contrast to the other species, and there are already molecular protocols for identification of *Ae. aegypti* and *Ae. albopictus,* we decided to molecularly identify this secondary vector to its species level. DNA extraction from 8 $F_0$ females of this species was carried out using a previously published protocol [40], and the quantities and qualities of the DNA in the samples was

determined using a Qubit 4 Fluorometer (Thermofisher Scientific) according to the manufacturer's instructions. The forward and reverse primers ITS1A: 5′-CCTTTGTACACACCGCCCGTCG-3′ and ITS1B: 5′-ATGTG TCCTGCAGTTCACA-3′ reported by [41] were used to amplify the internal transcribed spacer region 1 (ITS1) fragments of ribosomal RNA, with a fragment size of ~750 bp. The PCR was carried out using KAPATaq DNA polymerase in a total reaction volume of 15 µL. The reaction mix comprised of 1 µL of gDNA, 1. 5 µL of 10X Taq A Buffer, ~0. 4 µM (0.51 µL) each of forward and reverse primers, 1.25 mM (0.75 µL) of MgCl$_2$, 0.25 mM (0.15 µL) of dNTP mixes and 0.12 µL of Taq DNA polymerase (KAPA Biosystems, Wilmington, MA, USA) in ddH$_2$O. Amplification was carried out using the following conditions: initial denaturation of 5 min at 95 °C, followed by 35 cycles each of 30 s at 94 °C (denaturation), 40 s at 51 °C (primer annealing) and 60 s at 72 °C (extension). This was followed with 10 min final extension at 72 °C. The PCR amplicons were separated in a 1.5 % agarose gel stained with pEqGreen and visualized for bands. Eight PCR products were cleaned with a QIAquick® PCR Purification Kit (QIAGEN, Hilden, Germany) and sequenced on both strands, using the above primers by GenWiz, UK (https://www.genewiz.com/en-GB/). Forward and reverse sequence reads were aligned and analysed using the CLC Sequence Viewer version 6 [42] to obtain sequences. The sequences were checked using nucleotide BLAST (RRID:SCR_004870) [43] to search for similar sequences deposited by other researchers in the NCBI databases and to confirm their identity.

## WHO tube and CDC bottle insecticide susceptibility bioassays

Initially, the WHO tube test [44] were conducted with four insecticides: 0.03% deltamethrin, 0.03% alphacypermethrin, 0.25% permethrin, and 0.25% pirimiphos-methyl. For each assay, 4 replicates of 25 females (non-blood-fed, 3 to 5 d post-emergence) were used for the bioassays. After exposure to insecticide-impregnated papers, the mosquitoes were transferred to holding tubes, supplied with 10% sugar and the mortalities were recorded at 24 h. Alive and dead mosquitoes were stored for further analysis.

The CDC bottle bioassay [45] was also utilised to determine clothianidin and chlorfenapyr resistance. Mosquitoes from each population were exposed to 20 µg/bottle of clothianidin and 100 µg/bottle of chlorfenapyr. Four replicates of 25 females (non-blood-fed, 3 to 5 d post-emergence) were subjected to susceptibility tests according to the standard protocol [45]. After the diagnostic time of 60 minutes, the mosquitoes were transferred to holding cups covered with untreated nets and fed 10% sucrose. Mortality was recorded after 24 h for clothianidin, while daily records were taken for chlorphenapyr until after 72 h, when mortality was determined. Samples of survivors and dead mosquitoes were also stored in Eppendorf tubes at −20 °C for further analysis.

## Determination of metabolic resistance mechanisms

Biochemical assays for general esterases and monooxygenases were performed on exposed and unexposed cohorts of adult *Aedes albopictus* mosquitoes according to previously published methods [46] with some modifications. Adult female mosquitoes were individually homogenised in 200 µL of distilled water in a flat-bottomed microtitre plate on ice. The plates were spun at 3000 rpm and 4 °C for 15 min and the supernatants were used as the crude source of enzymes. For each assay, blank replicates (all components of the reaction mixture except for the enzyme source) were provided.

### Monooxygenases assay

The levels of P450 monooxygenases were determined using a previously established protocol [47]. The reaction mixture in each well of the microtitre plate contained 20 μL of insect homogenate, 80 μL of 0.0625 M potassium phosphate buffer pH 7.2, 200 μL of 3, 3′, 5, 5′ tetramethyl benzidine (TMBZ) solution (0.01 g TMBZ dissolved in 5 mL methanol plus 15 mL of 0.25 M sodium acetate buffer pH 5.0) and 25 μL of 3% hydrogen peroxide. The plates were incubated at room temperature for 2 h and absorbance was read at 450 nm as an endpoint in a Biotek ELX 808U plate reader. The values were compared with a standard curve of purified *cytochrome c* and were reported as equivalent units of cytochrome P450/mg protein corrected for the known content of *cytochrome c* and CYP450.

### General esterases

General esterase activity was determined using $a$-naphthyl acetate as a substrate. Reaction mixtures contained 20 μL of insect homogenate in microtitre plate wells (assigned $a$) and 200 μL of $a$-naphthyl acetate solution (120 μL of 30 mM $a$-naphthyl acetate dissolved in 12 mL 0.02 M phosphate buffer pH 7.2). The reaction mixtures were incubated at room temperature for 30 min before the addition of 50 μL of fast blue solution (0.023 g fast blue dissolved in 2.25 mL distilled water and 5.25 mL of 5% SDS, 0.1 M sodium phosphate buffer pH 7) to each well. Plates were incubated at room temperature for 5 min and the absorbance was read at 570 nm as an endpoint. The optical densities (OD) were compared with standard curves of OD for a known concentration of $a$-naphthol, to convert the absorbances to product concentration. The enzyme activities were reported as nmol of $a$-naphthol/min/mg protein.

## Data analysis, validation and quality control

The results of bioassays were tabulated in an Excel spreadsheet and the percentage mortalities calculated. Susceptibility to insecticides was recorded as mortalities within the ranges, 98–100%, mortality rates between 90 and 97% indicated possible resistance, while mortality below 90% confirmed the presence of insecticide resistant genes [44, 45].

Inferential statistics (independent sample *t*-tests) were used to compare the mean enzyme concentrations between exposed and unexposed mosquitoes. Statistical significance was determined at $p < 0.05$, and 95% confidence intervals (CIs) were reported. Enzyme concentrations were visualized using box plots. All analyses and visualizations were conducted using R (version 4.5.0, RRID:SCR_001905) [48].

## RESULTS

## Morphological identification of *Aedes* species

Morphological identification revealed the presence of four medically important *Aedes* species across the study sites and collection methods: *Ae. aegypti* (NCBI:txid7159), *Ae. albopictus* (NCBI:txid7160), *Ae. luteocephalus* (NCBI:txid299629) and *Ae. simpsoni* complex (NCBI:txid7161). Among these, *Ae. aegypti* was the most abundant species, followed by *Ae. albopictus*, while *Ae. luteocephalus* and *Ae. simpsoni* complex were less frequently encountered.

## Molecular identification of *Ae. luteocephalus*

Taking advantage of the primers previously created for container-breeding *Aedes* mosquitoes, fragments of the barcoding gene ITS1 were amplified and sequenced. 499-bp

fragments were retrieved from the sequencing and used for homology BLAST searches using NCBI. There were only two sequences of ITS1 for this species in NCBI, and the closest hit is a 664 bp fragment of Kenyan *Ae. luteocephalus* sequences, which share 99.39% identity with our sequence. This sequence with accession number KU056502 was deposited by [49] from a yet-to-be-published study. The three ITS1 sequences from this study have been deposited in GenBank with the accession numbers: PX377018–PX377020.

## Insecticide resistance profiles of the *Aedes* species

Bioassays established susceptibility to all three pyrethroids tested in all four species. *Aedes albopictus* showed complete susceptibility (100%) to all three pyrethroids (alphacypermethrin, deltamethrin and permethrin), in addition to full susceptibility towards clothianidin and chlorfenapyr. However, possible pirimiphos-methyl resistance was observed in Ndiezoke and Oyege communities (96% and 97% mortalities, respectively) and resistance (mortality = 87%) to DDT in Ndiezoke (Figure 2A). Similar trends were observed for *Ae. aegypti* with full susceptibility for all insecticides, including DDT (Figure 2B). *Aedes luteocephalus* was assayed only at the Ndiezoke community, and exhibited full susceptibility towards the three pyrethroids, chlorfenapyr and clothianidin, but recorded resistance (88% mortality) to pirimiphos-methyl (Figure 2C). Meanwhile, *Ae. simpsoni* complex mosquitoes which were assayed only in Oyege community were susceptible to all insecticides tested (Figure 2D).

## Determination of the potential role of metabolic resistance mechanism

Figure 3 shows the mean concentration of $\alpha$-esterase and monooxygenase enzymes in the exposed and unexposed mosquitoes (*Aedes albopictus*). For $\alpha$-esterase, exposed mosquitoes showed 3.4 times higher concentrations ($2.66 \times 10^{-5}$ µg/ml) than unexposed mosquitoes ($7.89 \times 10^{-6}$ µg/ml). However, this difference was not statistically significant, $t$ (8.01) = 1.95, $p$ = 0.087, 95% CI [ $-3.38 \times 10^{-6}$, $4.08 \times 10^{-5}$]. Similarly, the concentration of monooxygenase was 2.54 times higher (2.945970 µg/ml) in exposed mosquitoes compared to unexposed mosquitoes (1.159013 µg/ml). However, this difference was not statistically significant, $t$ (10.77) = 1.72, $p$ = 0.114, 95% CI [−0.50, 4.08]. Although the data suggest a potential trend toward higher enzyme concentrations in exposed mosquitoes, the result does not provide conclusive evidence of a statistically significant effect of insecticide exposure linked spike in enzyme levels.

Analysis of the relative enzyme abundance (REA), a function of consumed substrate per unit time, revealed higher levels of both enzymes in insecticide-exposed *Ae. albopictus*, compared to their unexposed counterparts. For $\alpha$-esterase, exposed mosquitoes showed a mean REA of $3.30 \times 10^{-9}$ mol/hour/mg, whereas unexposed mosquitoes exhibited a lower mean REA of $5.78 \times 10^{-10}$ mol/hour/mg. Similarly, monooxygenase abundance was higher in exposed mosquitoes ($1.55 \times 10^{-6}$ mol/hour/mg) compared to unexposed mosquitoes ($1.13 \times 10^{-6}$ mol/hour/mg).

## Data validation and quality control

All data reported here has been curated, and the terminology has been harmonized before being published by the National Arbovirus and Vectors Research Centre (NAVRC) in GBIF. Data has been validated using the GBIF IPT (Integrated Publishing Toolkit) validator tool.

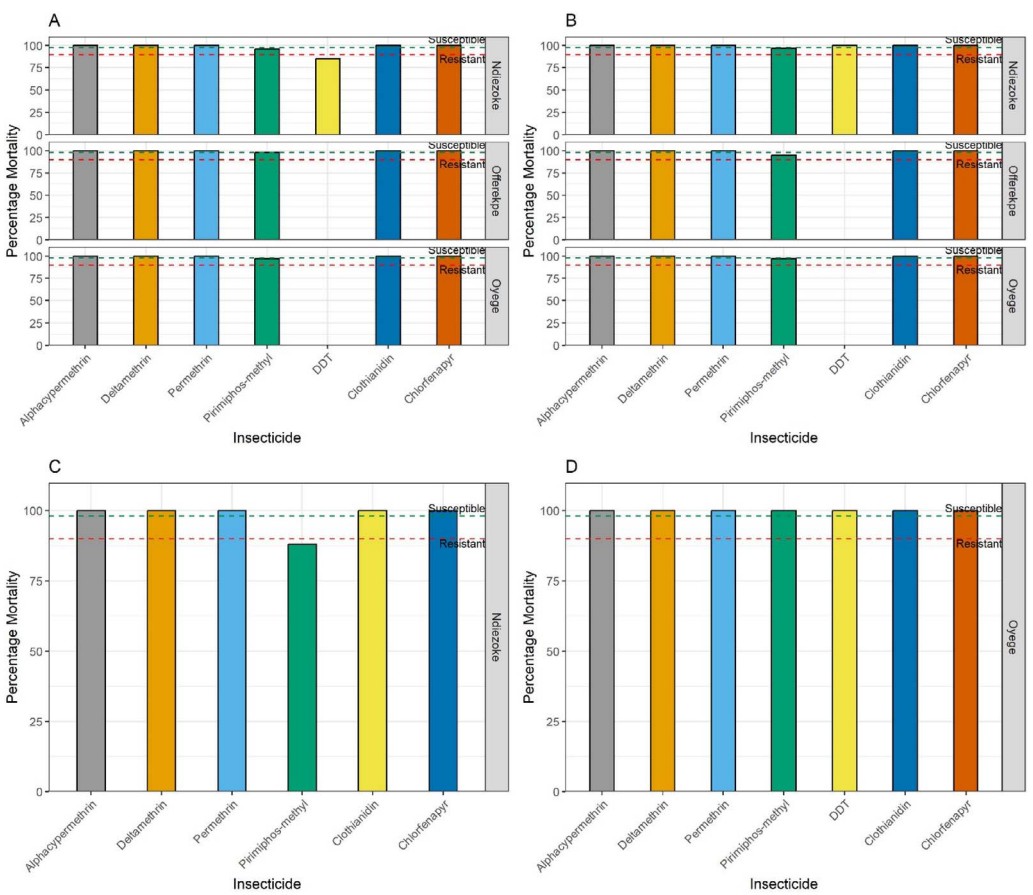

**Figure 2. Insecticide resistance profiles of the various *Aedes* populations.**
Each bar represents the percentage mortalities for insecticides tested. (A) for *Ae. albopictus*, (B) for *Ae. aegypti*, (C) for *Ae. luteocephalus* and (D) for *Ae. simpsoni* complex. The green, dashed lines (≥98% mortality) indicate full susceptibility to insecticide, while the red, dashed lines (<90%) indicate resistance.

## DISCUSSION

This study examined the composition and resistance profiles of *Aedes* mosquitoes mostly breeding in sympatry in rural communities of Southeastern Nigeria. *Aedes* species are distributed across diverse ecological zones in the country, with the wetter and more humid southern regions generally supporting greater species diversity than the northern areas [12, 28, 30, 50]. Evidence from some of these studies indicates that in the drier northern zones, *Ae. aegypti* is usually the predominant and in many cases, the only *Aedes* species reported.

Insecticide bioassays revealed substantial susceptibility towards public health insecticides from several classes. Notably, to our knowledge, this study provides the first global report on the insecticide resistance profile of *Ae. luteocephalus*, which was found to be susceptible to all insecticides tested except pirimiphos-methyl. This finding underscores the limited exposure of this sylvatic vector to chemical interventions and highlights the importance of expanding resistance monitoring beyond the relatively well-studied *Ae. aegypti* and *Ae. albopictus*. The substantial susceptibility observed in these *Aedes* species aligns with previous studies conducted in several sites in Nigeria and other regions. For instance [51], found that *Ae. aegypti* in Lagos, Southwestern Nigeria, was susceptible to

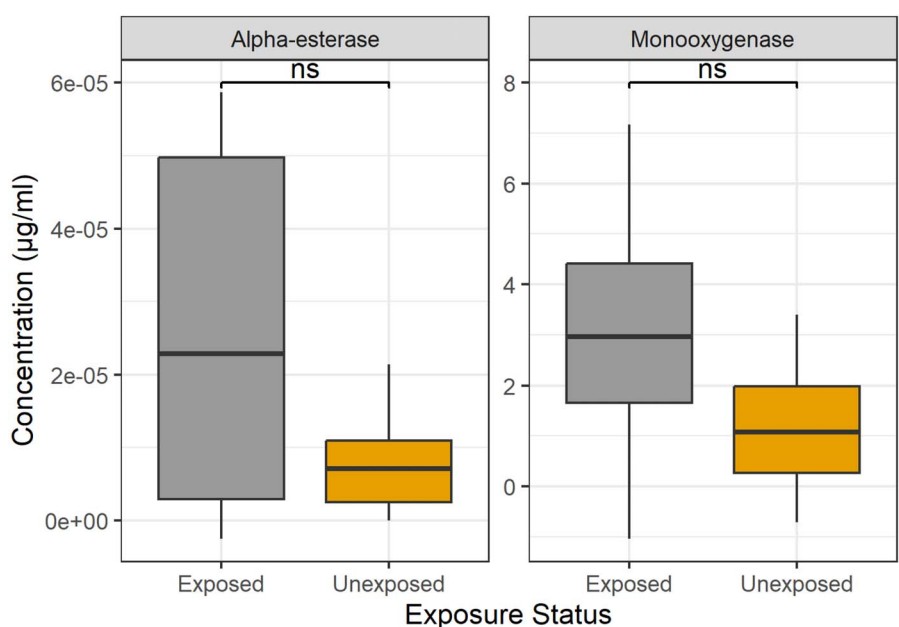

**Figure 3.** Mean concentration of enzymes in the exposed and unexposed *Ae. albopictus*. Statistical analysis showed no significant difference (ns).

deltamethrin and permethrin across various sites, though resistance to DDT was noted. However, in another study in other parts of Lagos, high-level *Ae. aegypti* resistance to DDT and pyrethroid was recorded [31]. Similar to our findings [52], reported that *Ae. aegypti* populations in Abia State, Southeastern Nigeria, were susceptible to all insecticides tested except for DDT. Also, in Kwara State, North-central, Nigeria [36], reported complete susceptibility of *Ae. aegypti* to permethrin and DDT. However, contrasting patterns were observed in far Northern Nigeria, where resistance to several pyrethroid insecticides and a significantly high level of DDT resistance in *Ae. Aegypti* have previously been reported [12]. Further evidence from other regions shows varying levels of resistance: Nwankwo *et al.* [53] found that *Ae. aegypti* in Anambra State, Southeastern Nigeria, was susceptible to deltamethrin but resistant to DDT, while *Ae. albopictus* showed possible resistance to deltamethrin and resistance to DDT.

Elsewhere in Africa, Konkon *et al.* [54] has found widespread pyrethroid resistance in *Ae. aegypti* across their study sites in Benin, although *Ae. albopictus* populations from the same locations remained susceptible to the pyrethroid insecticides. In Burkina Faso, Badolo *et al.* [22] recorded *Ae. aegypti* resistance to pyrethroids (deltamethrin and permethrin) and bendiocarb, though the mosquitoes were susceptible to organophosphates (fenithrothion and malathion). In Cameroon, Djiappi-Tchamen *et al.* [55] reported DDT resistance in both *Ae. albopictus* and *Ae. aegypti*, and pyrethroid resistance in *Ae. aegypti* in some locations, although populations in other sites remained largely susceptible to several insecticides. Contrary to this, a study in Tanzania indicated that *Ae. aegypti* was largely susceptible to commonly used insecticides [56], highlighting the regional variability in resistance patterns. In Asia, a nationwide study in Malaysia by Ishak *et al.* [57] reported widespread DDT resistance in *Ae. aegypti*, while pyrethroid-resistant *Ae. aegypti* populations have been documented in Vietnam [58].

The low level of resistance observed in the study area could be attributed to the specific ecological and behavioural characteristics of these *Aedes* species. These mosquitoes are typically outdoor biters and tend to avoid breeding in environments contaminated with xenobiotics, such as farm puddles or household ditches, which are often sources of insecticide exposure [9, 59]. As a result, their exposure to insecticide residues is likely limited, particularly because there is no outdoor vector control intervention in the study area, which may explain their continued susceptibility. The findings from our study are consistent with those from similar environmental settings but differ from regions where more intensive agricultural or urban insecticide use has led to higher resistance levels.

Our findings demonstrate a consistent pattern in which both P450 monoxygenase and esterase enzyme concentration and relative enzyme abundance (REA) are higher in insecticide-exposed *Ae. albopictus* compared to their unexposed counterparts. Although the statistical tests for enzyme concentrations did not reveal significant differences ($p = 0.087$ for $\alpha$-esterase and $p = 0.114$ for monooxygenase), the descriptive data suggest a biologically meaningful trend toward increased enzyme levels in response to insecticide exposure. Such a response may reflect metabolic adaptation, whereby mosquitoes enhance detoxification pathways to counteract insecticidal effects. This finding aligns with the study by Fagbohun *et al.* in Lagos, Nigeria [31], which reported elevated monooxygenase activity in *Ae. aegypti*, although this was inversely correlated with mortality rates in response to DDT and permethrin. Similar resistance mechanisms have also been reported in *Ae. aegypti* in Côte d'Ivoire [60], where resistance to pyrethroids, DDT, and organophosphates was linked to increased alpha-esterase activity. Further corroborating these findings, in the Central African Republic, documented elevated levels of esterases and monooxygenases in *Ae. aegypti* and *Ae. albopictus* populations resistant to pyrethroid insecticides and DDT have been documented [61]. This enzymatic activity is crucial for detoxifying insecticides and may contribute to the gradual development of resistance. Rodríguez *et al.* [62] have also emphasized the role of mixed-function oxidases and esterases in organophosphate resistance in *Ae. aegypti* populations in Latin America. In a more recent finding it has been reported that the pyrethroid resistance of *Ae. albopictus* in urban and rural areas of Cambodia was largely due to monooxygenases [19].

Additionally, studies in other parts of Africa [63, 64] and in Europe [64] have associated elevated levels of esterases, GSTs, or mixed-function oxidases with resistance to pyrethroids, carbamates and organophosphates in *Ae. aegypti* populations.

Both RNA sequencing (RNA-seq) and DNA microarray approaches have been widely applied to investigate the role of cytochrome P450 monooxygenases (CYP450s) in *Aedes* insecticide resistance. In Malaysian populations of *Ae. albopictus,* microarray-based transcription profiling revealed that metabolic resistance through CYP450 up-regulation contributed to pyrethroid resistance [57]. Notably, *CYP6P12* over-expression was strongly associated with this resistance. Similarly, using RNA-seq and targeted DNA-seq on resistant *Ae. aegypti* from multiple continents, over-transcription of numerous detoxification genes has been reported [65]. This reported resistance including members of the *CYP9J* subfamily, several of which were functionally linked to deltamethrin metabolism. In Vietnam, transcriptome sequencing of *Ae. aegypti* resistant to DDT, carbamates, and pyrethroids identified highly upregulated CYP450s (up to 8.3-fold) and glutathione-S-transferase (GST) genes (up to 3.5-fold) [66]. Collectively, these findings reinforce the critical role of metabolic

resistance mechanisms, particularly the over-expression of CYP450s in conferring multi-class insecticide resistance in *Aedes* populations across diverse ecological settings.

The reduced sensitivity of *Ae. aegypti*, *Ae. albopictus*, and *Ae. luteocephalus* to pirimiphos-methyl observed in our study could be linked to these underlying biochemical mechanisms. These findings indicate that while resistance remains relatively low in our study area, there are detectable mechanisms that could potentially drive resistance under different environmental conditions or selective pressures.

DNA barcoding and PCR-based species identification have become indispensable tools in accurately identifying mosquito vectors of public health importance, especially when morphological similarities or cryptic species limit traditional identification [67, 68]. Markers such as ITS1, ITS2, and COXI have proven effective in distinguishing species like *Ae. aegypti* and *Ae. albopictus* [41, 69, 70]. However, the absence of molecular tools for species like *Ae. luteocephalus* highlights existing gaps in our understanding. In this study, combining morphological identification with molecular analysis allowed for robust confirmation of *Ae. luteocephalus*, reinforcing the need for developing targeted molecular diagnostics for all vector species. Accurate species identification is crucial for informing control strategies in regions where multiple *Aedes* species coexist and contribute to the transmission of arboviral diseases, such as dengue, chikungunya, Zika, and yellow fever.

Our study has some limitations. Firstly, the collections were limited to rural areas in only three communities, which may not fully represent the diversity of *Aedes* populations across the entire region. Secondly, only biochemical assays were used to investigate metabolic resistance mechanisms, while molecular methods such as RNA-seq or microarray profiling could have provided more detailed insights. Thirdly, due to the low number of *Aedes* female-survivors in the bioassays, we were unable to genotype resistant and susceptible samples for kdr mutations. Nonetheless, the data presented here offer important baseline information on *Aedes* resistance profiles in southeastern Nigeria and provides a foundation for future studies.

## CONCLUSION

The observed variability in insecticide resistance among *Aedes* species across different geographical regions underscores the importance of localized monitoring and management strategies in vector control. Our findings suggest that the intensity of insecticide resistance in *Aedes* mosquitoes across West Africa is still relatively low, especially when compared to *Anopheles* mosquitoes, which are known to exhibit higher resistance levels. However, this could change with shifts in environmental conditions, agricultural practices, or public health interventions. Continuous surveillance and research are essential to ensure that insecticide-based control measures remain effective. By understanding and addressing the specific factors influencing resistance in each region, public health authorities can develop more targeted and adaptive strategies. This approach is crucial for maintaining the efficacy of vector control programmes and mitigating the risk of vector-borne diseases such as yellow fever, dengue, Zika and chikungunya, which are primarily transmitted by *Aedes* mosquitoes. The ongoing monitoring of resistance patterns and mechanisms will play a vital role in informing the selection of insecticides and the design of integrated vector management programmes tailored to local conditions, ultimately contributing to better health outcomes in affected communities.

## DATA AVAILABILITY

The data supporting this article are published through the Integrated Publishing Toolkit (IPT) of GBIF Africa and are available under a CC0 waiver [71].

The gDNA sequences of *Ae. luteocephalus* ITS1 have been deposited into GenBank, with accession numbers: PX377018–PX377020.

## EDITORS' NOTE

This paper is part of a series of Data Release articles working with GBIF and supported by TDR, the Special Programme for Research and Training in Tropical Diseases hosted at the World Health Organization, in order to publish datasets on vectors of human diseases [72].

## LIST OF ABBREVIATIONS

BLAST: Basic Local Alignment Search Tool; CDC: Centers for Disease Control and Prevention; CI: Confidence Interval; COX: Cytochrome Oxidase; CYP: Cytochrome P450; DDT: Dichlorodiphenyltrichloroethane; DNA: Deoxyribonucleic Acid; GBIF: Global Biodiversity Information Facility; GST: Glutathione S-transferase; IPT: Integrated Publishing Toolkit; ITS: Internal Transcribed Spacers; LGA: Local Government Area; NAVRC: National Arbovirus and Vectors Research Centre; NCBI: National Center for Biotechnology Information; NCDC: Nigeria Centre for Disease Control; OD: Optical Density; PCR: Polymerase Chain Reaction; REA: Relative Enzyme Activity; RNA-Seq: Ribonucleic Sequencing; WHO: World Health Organization.

## DECLARATIONS

### Consent to participate

Community approval was obtained before the commencement of the study. Written informed consent was obtained from all mosquito collectors and supervisors. All field staff received yellow fever vaccination before participation. In cases where collectors were diagnosed with malaria during the study, they were treated free of charge in accordance with World Health Organization (WHO) recommendations. Participation was entirely voluntary and individuals were free to withdraw from the study at any time without any consequences.

### Competing interests

The authors declare that they have no competing interests.

### Authors' contributions

UCN: Conceptualization, resources, methodology, investigation, supervision, data curation, project administration, writing (original draft); POU: Supervision, methodology, project administration, writing (review and editing); PCO: Software, data curation, methodology, visualization, writing (review and editing); MMM: Investigation, methodology, writing (review and editing); CMN: Investigation, methodology, writing (review and editing); IMN: Investigation, methodology, writing (review and editing); ONN: Investigation, methodology, writing (review and editing); SOA: Investigation, methodology, writing (review and editing); LCI:  Investigation, data curation, methodology, writing (review and editing); JEO: Investigation, methodology, writing (review and editing); NOA: Investigation, methodology, writing (review and editing); CKE: Investigation, methodology, writing

(review and editing); FAD: Investigation, methodology, writing (review and editing); COO: Investigation, methodology, writing (review and editing); EIE: Investigation, methodology, writing (review and editing); IUI: Investigation, methodology, writing (review and editing); EON: Supervision, methodology, writing (review and editing); SCN: Investigation, methodology, writing (review and editing); CUN: Investigation, methodology, writing (review and editing); JUA: Software, data curation, visualization, writing (review and editing); RAO: Investigation, methodology, writing (review and editing); AS: Investigation, methodology, writing (review and editing); FRA: Investigation, methodology, writing (review and editing); CA: Resources, project administration, writing (review and editing); OCC: Supervision, methodology, writing (review and editing); SSI: Conception, Resources, methodology, investigation, supervision, project administration, writing (review and editing).

## Funding

The authors had no specific funding for this work.

## Acknowledgements

We are grateful to all the community stakeholders for their warm reception and support throughout the project. We also extend our sincere appreciation to the mosquito collectors whose efforts made the fieldwork a success. Our gratitude goes to the staff and management of the National Arbovirus and Vectors Research Centre (NAVRC) for their various forms of support. We acknowledge Tsiky Robetrano (GBIF Africa) for his assistance with the use of the IPT in publishing our data in GBIF and Paloma Helena Fernandes Shimabukuro for emphasizing the importance of publishing vector-based studies in GBIF.

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
