## [Editor Report]

Editor’s AssessmentIn Africa, arbovirus outbreaks are increasing in frequency and intensity. Aedes being principal vectors of the arboviruses that cause yellow fever, chikungunya and dengue in the human population. However systematic surveillance data on these species remains limited, hindering for entomological and modelling research and control strategies. This paper is one of a series of Data Release papers in GigaByte supported by TDR and the WHO describing datasets hosted in GBIF to tackle these data gaps in vectors of human disease data. This paper this investigated the composition and resistance profiles of Aedes species in rural areas of south-eastern Nigeria. Identifying them to species level of the four main Aedes species, and looking at their resistance towards public health insecticides (which worryingly was established to be very low). Peer review and data auditing found the data to be well validated (which also included DNA barcoding). This work and the extremely useful data provided representing an important step towards building a pan-African resource for Aedes mosquito data collection.Editor’s AssessmentIn Africa, arbovirus outbreaks are increasing in frequency and intensity. Aedes being principal vectors of the arboviruses that cause yellow fever, chikungunya and dengue in the human population. However systematic surveillance data on these species remains limited, hindering for entomological and modelling research and control strategies. This paper is one of a series of Data Release papers in GigaByte supported by TDR and the WHO describing datasets hosted in GBIF to tackle these data gaps in vectors of human disease data. This paper this investigated the composition and resistance profiles of Aedes species in rural areas of south-eastern Nigeria. Identifying them to species level of the four main Aedes species, and looking at their resistance towards public health insecticides (which worryingly was established to be very low). Peer review and data auditing found the data to be well validated (which also included DNA barcoding). This work and the extremely useful data provided representing an important step towards building a pan-African resource for Aedes mosquito data collection.

---

## [Reviewer Report]

Upload additional filesDRR-202508-03-R01/stage_files/DRR-202508-03/Review MS/DRR-202508-03_Data-review-BM.pdfReviewer name and names of any other individual's who aided in reviewer Bastien MolcretteDo you understand and agree to our policy of having open and named reviews, and having your review included with the published papers. (If no, please inform the editor that you cannot review this manuscript.)YesIs the language of sufficient quality?YesPlease add additional comments on language quality to clarify if needed
Are all data available and do they match the descriptions in the paper? NoAdditional CommentsGBIF dataset is OK and matches the descriptions in the paper; but the DNA barcode data are not accessible from NCBI; they should be checked before the manuscript is formally accepted.Are the data and metadata consistent with relevant minimum information or reporting standards? See GigaDB checklists for examples <a href="http://gigadb.org/site/guide" target="_blank">http://gigadb.org/site/guide</a>YesAdditional CommentsIs the data acquisition clear, complete and methodologically sound?YesAdditional CommentsIs there sufficient detail in the methods and data-processing steps to allow reproduction?YesAdditional CommentsIs there sufficient data validation and statistical analyses of data quality? YesAdditional CommentsIs the validation suitable for this type of data?YesAdditional CommentsIs there sufficient information for others to reuse this dataset or integrate it with other data?YesAdditional CommentsAny Additional Overall Comments to the Authoronly minor comments, see the additional fileRecommendationMinor Revision

---

## [Reviewer Report]

Reviewer name and names of any other individual's who aided in reviewer Samia BOUSSAADo you understand and agree to our policy of having open and named reviews, and having your review included with the published papers. (If no, please inform the editor that you cannot review this manuscript.)YesIs the language of sufficient quality?YesPlease add additional comments on language quality to clarify if needed
Are all data available and do they match the descriptions in the paper? YesAdditional CommentsAre the data and metadata consistent with relevant minimum information or reporting standards? See GigaDB checklists for examples <a href="http://gigadb.org/site/guide" target="_blank">http://gigadb.org/site/guide</a>YesAdditional CommentsIs the data acquisition clear, complete and methodologically sound?YesAdditional CommentsIs there sufficient detail in the methods and data-processing steps to allow reproduction?YesAdditional CommentsIs there sufficient data validation and statistical analyses of data quality? Not my area of expertiseAdditional CommentsIs the validation suitable for this type of data?YesAdditional CommentsIs there sufficient information for others to reuse this dataset or integrate it with other data?YesAdditional CommentsAny Additional Overall Comments to the AuthorRecommendationAccept